# Variation in Hemodynamic Characteristics during Periodontal Crown-Lengthening Surgical Procedure: An Uncontrolled Cohort Study

**DOI:** 10.3390/healthcare10050919

**Published:** 2022-05-16

**Authors:** Abdullah Saad Alqahtani, Rajashekhara Bhari Sharanesha, Khalid Gufran, Nasser Raqe Alqhtani, Alwaleed Abushanan, Mohammed Alasqah, Abdulaziz Mohammad Alsakr, Hassan Alkharaan

**Affiliations:** 1Department of Preventive Dental Sciences, College of Dentistry, Prince Sattam Bin Abdulaziz University, Alkharj 11942, Saudi Arabia; ab.alkahtani@psau.edu.sa (A.S.A.); r.bhari@psau.edu.sa (R.B.S.); a.abushanan@psau.edu.sa (A.A.); m.alasqah@psau.edu.sa (M.A.); a.alsakr@psau.edu.sa (A.M.A.); h.alkharaan@psau.edu.sa (H.A.); 2Department of Oral and Maxillofacial Surgery and Diagnostic Science, College of Dentistry, Prince Sattam Bin Abdulaziz University, Alkharj 11942, Saudi Arabia; n.alqhtani@psau.edu.sa

**Keywords:** periodontal crown lengthening, hemodynamic parameters, oxygen saturation

## Abstract

(1) Background: The purpose of this prospective study was to determine the changes in primary hemodynamic parameters and oxygen saturation in systemically healthy patients during the surgical procedure involving crown lengthening. (2) Methods: A total of 44 patients who required a crown-lengthening procedure in a single tooth in the maxillary arch were included in this study. Heart rate (HR), blood pressure (BP) and oxygen saturation (SpO_2_) were measured in all the subjects at three different intervals: before injecting the anesthetic (T1), after the anesthetic injection (T2) and after the procedure (T3). Descriptive statistics were computed, and observations were recorded as mean and standard deviation (SD). Analysis of variance (ANOVA) was used to compare the mean observation within parameters at different time intervals. (3) Results: All primary hemodynamic parameters were increased in the T2 phase over T1 and decreased in the T3 phase over T2. However, SpO_2_ decreased in both the T2 and T3 phases compared to the initial T1 phase. No significant differences were observed among the primary hemodynamic variables. However, SpO_2_ showed a significant difference (*p* = 0.013) among the T1, T2 and T3 phases. (4) Conclusions: Further study with larger sample size is required in order to analyze the accurate hemodynamic alterations.

## 1. Introduction

The use of a local anesthetic agent is inevitable in a dental surgical procedure. The local anesthesia used in dentistry usually includes lidocaine, mepivacaine, prilocaine and articaine. These anesthetic drugs are used with vasoconstrictors for slowing down the absorption of the anesthetic drug, thus prolonging the action of anesthetic agents [1]. This vasoconstrictor increases the safety level as well as maintains ischemia up to a certain level, which lessens the bleeding during surgical procedures [2]. However, the presence of a vasoconstrictor in the local anesthetic agent may cause hemodynamic variations, which influence other factors in patients such as anxiety or stress [3].

There are a few complications of dental anesthesia recorded in previous studies such as syncope followed by laryngospasm, phlebitis, dysrhythmia, bronchospasm, and hypotension [4,5,6]. Death was also recorded as one of the most crucial side effects of anesthesia in dentistry, caused by hypoxia characterized by low blood pressure, and a low cardiac output with severe bradycardia [7]. Nevertheless, the elevation of blood pressure during surgery is more related to patients’ anxiety and not associated with the vasoconstrictor in the anesthetic agent [8].

Stress is the reaction of the body to threat, while anxiety is the response of the body to stress. This causes an alteration in the respiratory rate, which is proportional to altered oxygen saturation and/or carbon dioxide levels in the blood [9]. According to Agras et al., (1969), dental anxiety is the fifth most common cause of anxiety. It is an emotional state due to a threatening stimulus that may not be identifiable. A reaction to known or perceived danger or threat is fear. Both dental anxiety and stress evoke physical, cognitive, emotional, and behavioral responses in an individual [10]. Many factors such as past dental experience, the dental experience of friends or relatives, and congenital aspects determine the anxiety or fear of dental treatment among dental patients [11,12].

Surgical management of periodontal disease includes the periodontal flap operation, grafts and gingivectomy, which are carried out under local anesthesia. It is known that an alteration in blood pressure during the dental surgical procedure is natural even in physically healthy patients [13]. Many studies have assessed the prevalence and etiology of dental anxiety in distinctive age groups [14,15,16,17], and a few studies have evaluated dental anxiety and vital signs during different dental procedures [18,19,20]. Hence, it becomes necessary to understand the hemodynamic alterations such as systolic and diastolic blood pressure as well as oxygen saturation during a dental surgical procedure involving crown lengthening in systemically healthy individuals.

## 2. Materials and Methods

The research committee of the College of Dentistry, Prince Sattam bin Abdul Aziz University approved this study. Participants were recruited from the department of periodontology, college of dentistry, Prince Sattam bin Abdul Aziz University, Saudi Arabia. Patients who were being referred to the department of periodontics for crown-lengthening procedures were selected for this study.

For patients to be included in this study, the following inclusion criteria were followed: all the participants were above 20 years of age, and subjects were free of any systemic disease that contra-indicates periodontal surgery or can influence the HR or BP. On the contrary, pregnant or lactating patients and patients allergic or sensitive to local anesthetics, patients on medication that affects the HR or BP or on nerve-blocking agents, extremely apprehensive patients, or anemic patients were also excluded from this study. In addition, patients with calloused fingers or with artificial nails or nail polish were not included in the study either. Informed consent was obtained from each subject after explaining a brief introduction of the study. 

Physiological changes in HR, BP and SpO_2_ were measured in all the subjects at three different intervals such as before injecting the anesthetic (T1), after the anesthetic injection (T2) and after the procedure (T3). A single calibrated examiner recorded all the readings from all the participants. 

A total of 130 patients were screened for this study and 44 patients fulfilled the study criteria to be included in this study. A total of 24 male (mean age 38.08 ± 7.98 years) and 20 female (mean age 36.41 ± 5.18) were selected for this research. The selected individuals required a crown-lengthening procedure in the maxillary arch on either the right or left side ofa single tooth. Only one cartridge of anesthesia with epinephrine (1:80,000) (DarouPakhshCompany, Tehran, Iran) and lidocaine (2%) was given to all participants. If any participants required more than one cartridge of anesthesia, then those participants were excluded from this study. A pulse oximeter (Gibson Company, Hamburg, Germany) was used to assess the SpO_2_ by attaching it to the patient’s finger. In addition, an automatic sphygmomanometer (Omron Healthcare, Lake Forest, IL, USA) was used to measure the hemodynamic alterations. After assessing the oral cavity, treatment was initiated in one dental arch only. A single dentist carried out the treatment of the selected patients at the same time every day (10 a.m.) and the procedure lasted about 45 min.

Subjects were examined in an upright position in a dental chair. The left index finger of the patient was inserted into the pulse oximeter until it touched the sensor, after which it was switched on. The pulse oximeter was disinfected using 70% isopropyl alcohol before use. All the findings were separately noted in an excel sheet and the readings were subjected to statistical analysis.

Descriptive statistics were computed, and observations were recorded as mean and standard deviation (SD). Analysis of variance (ANOVA) was used to compare the mean observation within parameters at different intervals. All statistics were computed at a 95% confidence interval (CI) and a *p*-value ≤0.05 was selected as statistically significant. All the statistical analyses were performed with SPSS, Version 27 (IBM Co., Armonk, NY, USA) software.

## 3. Results

Table 1 shows the mean hemodynamic values observed in the T1, T2 and T3 phases. At the T1 phase, the mean systolic blood pressure observed was 134.82 ± 8.45 mmHg, and Diastolic blood pressure was 79.53 ± 9.32 mmHg. The mean heart rate was 76.56 ± 7.11 BPM and oxygen saturation was 97.21 ± 1.21%. In the T2 phase, there was a slight increase in systolic and diastolic blood pressure to 135.34 ± 6.24 mmHg and 81.04 ± 7.01 mmHg, respectively. HR also showed an increase from 76.56 ± 7.11 BPM to 79.46 ± 8.18 BPM. Oxygen saturation was at 97.21 ± 1.21% before administration of LA, which decreased to 97.0 ± 1.24 after administration of LA. After the surgical procedure, the systolic blood pressure decreased to 133.12 ± 4.89, and the diastolic blood pressure slightly increased to 80.80 ± 5.06. Heart rate slightly decreased to 78.39 ± 7.46 and oxygen saturation decreased to 96.43 ±1.18% (Figure 1).

ANOVA showed no significant difference among T1, T2 and T3 phases in BP (systolic BP, *p* = 0.294 and diastolic BP, *p* = 0.607) and HR (*p* = 0.221). However, only SpO_2_ exhibited a statistically significant difference in the ANOVA test (*p* = 0.013) (Table 2).

## 4. Discussion

This study evaluated the hemodynamic changes in patients with a surgical crown-lengthening procedure. In dentistry, several factors play an imperative role in changing hemodynamic parameters during treatment such as the stress associated with physical and psychological responses and a previous episode of pain [21]. The endogenous release of adrenaline, which is a potent factor in causing cardiovascular change, could be influenced by an episode of anxiety and pain associated with therapeutic dental procedures [9]. It is also vital to consider the effects of high-dose administration of local anesthetic agents, which could cause systemic changes mainly targeting the CVS andCNS systems. Lidocaine and epinephrine are most widely used as a combination in local anesthesia, where an increased epinephrine ratio allows better command over vascular flow, which might cause transient enhancement of cardiovascular modification [22,23].

Fukayama et al., (2006) stated that certain hemo-modulating drugs and local anesthesia (LA) used in dental surgical procedures could lead to vascular variation. Along with the surgical procedures, additional factors like anxiety and stress may also contribute to the hemodynamic changes. Various studies have hypothesized that anxiety in patients towards dental procedures plays a major role in hemodynamic alterations [3,8]. Reasons such as a previous traumatic episode to oneself or to a family member, which has been exaggerated for various reasons, or any kind of myth circulating in society about dental procedures may play a crucial role in developing dental anxiety [11,24]. Furthermore, the complex appearance of dental instruments and procedures such asthe introduction of local anesthetic injections, the use of rotary instruments and surgical forceps might enhance the patient’s anxiety [25,26].

The present study was designed to assess changes in HR, blood pressure and oxygen saturation in patients before, during and after crown-lengthening surgical procedures. The results indicated a slight increase in both systolic and diastolic blood pressure after the introduction of the LA agent, which were found to be reduced post-procedure at133 and 80 mmHg, respectively. However, the noted difference in blood pressure is not statistically significant. Similar changes were noted with heart rate also showing an initial increase with the introduction of LA and reverting after the completion of the procedure. Statistically, a significant difference was noted with oxygen-saturation recordings, where it appeared to drop after the introduction of LA and further dropped to an extent after the surgical procedure.

HR and BP were generally increased during the dental treatment. This increase in BP and HR were contributed by the sympathetic nervous system, which is not clearly understood [21,27,28,29]. This is due to the fact that, other than during a dental procedure, LA does not elevate the plasma noradrenaline concentrations; therefore, it is assumed that the increase in BP is due to the dental treatment [27]. Previous studies witnessed that the administration of LA with adrenaline immediately increased the plasma noradrenaline concentration. Adrenaline containing local anesthetic agent outflows into the systemic circulation, unlike the activation of the sympathetic nervous system, which is the main reason for the increasing BP during dental treatment [21,27,28,29,30]. Moreover, after administrating LA, HR increased due to the excessive production of adrenaline hormone by the adrenal medulla [9].

Cardiovascular changes during dental treatment are distinctive with and without the use of vasoconstriction in a local anesthetic agent. The mean HR and BP are usually increased before injecting the LA; however, the HR decreases during the injection [31]. Likewise, injecting isotonic saline exhibits similar cardiovascular changes [31]. Meanwhile, LA with vasoconstrictor causes a 2–10 bpm higher HR than plain lidocaine [32]. The HR becomes more distinct and persistent when a higher concentration of the vasoconstrictors is used in the LA [32,33]. Similarly, LA without a vasoconstrictor does not alter the BP; however, LA with a vasoconstrictor shows a higher BP (0 to +8 mmHg), which increases more with the higher concentration of the vasoconstrictor present in the LA [31,32].

Matsumura et al., (1988) noted that changes in the hemodynamic status with alteration in BP and HR occur during dental surgical procedures [21], which is in agreement with the findings of the present study, whereas another study by Faraco et al., (2007) revealed no significant alteration in hemodynamics [23]. On the other hand, Gedik et al., (2005) reported a decrease in BP and HR, which contrasts with the findings of the present study [34]. Moreover, no significant change in mean SpO_2_ measurement was reported, indicating no significant hypoxic threat to the patient [23].

The variability noted in the cardiovascular balance during the dental surgical procedures can be attributed to the nature of the study, the different local anesthetic agents used, and the varied approaches to the hypothesis and types of surgical procedures. In addition, there is a need to educate patients and to use appropriate definite pharmacological or non-pharmacological means to ease an individual’s modulate cardiovascular balance.

Every study poses some limitations, and this study is not exceptional. This study selected all the samples based on the convenience sampling technique. A calculated sample size would have strengthened this study. In addition, no control group was determined in this study as all the patients who required crown lengthening and fulfilled the inclusion criteria were selected for the study group. A potential control group could have provided some valuable insight.

## 5. Conclusions

Distinctive dental treatment may have different effects on patients in terms of hemodynamic alteration. Regarding the current study, SpO_2_ may be associated with the anesthesia during periodontal surgery; however, other vital-sign alterations, HR and BP have no relation to the anesthesia used during periodontal surgery, including the crown-lengthening procedure. This conclusion might be strengthened by the presence of a control group. Therefore, more studies with a larger sample size and appropriate control group in different dental treatments would be required in order to observe the hemodynamic parameters.

## Figures and Tables

**Figure 1 healthcare-10-00919-f001:**
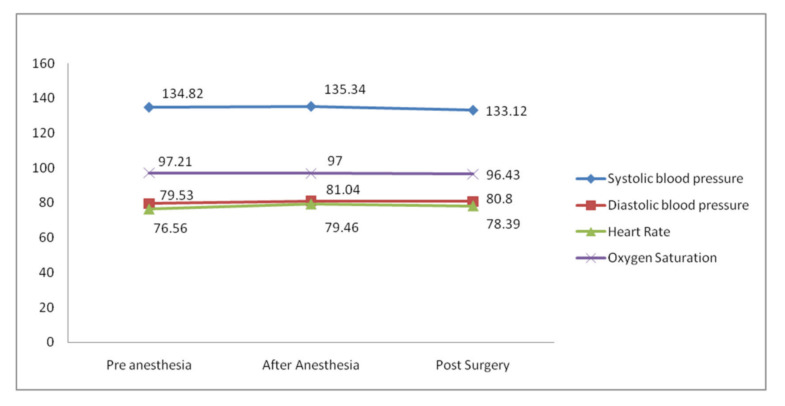
Hemodynamic characteristics were observed at different intervals.

**Table 1 healthcare-10-00919-t001:** Hemodynamic parameters were observed in the study.

Parameters	Interval	Mean	SD	Minimum	Maximum
Systolic blood pressure (mmHg)	T1	134.82	8.45	122.00	152.00
	T2	135.34	6.24	130.00	156.00
	T3	133.12	4.89	128.00	150.00
Diastolic blood pressure (mmHg)	T1	79.53	9.32	64.00	92.00
	T2	81.04	7.01	70.00	98.00
	T3	80.80	5.06	72.00	92.00
Heart Rate (BPM)	T1	76.56	7.11	67.00	88.00
	T2	79.46	8.18	67.00	99.00
	T3	78.39	7.46	67.00	92.00
Oxygen Saturation (%)	T1	97.21	1.21	95.00	99.00
	T2	97.00	1.24	95.00	98.00
	T3	96.43	1.18	95.00	98.00

SD, standard deviation; T1, before injecting the anesthetic; T2, after the anesthetic injection; T3, injecting after the procedure; MMHg, millimeters of mercury; BPM, Beats per minute; %, percentage.

**Table 2 healthcare-10-00919-t002:** ANOVA statistics.

Parameters	Sum of Squares	Df	Mean Square	F	*p*
Systolic blood pressure (mmHg)	110.748	2	55.374	1.236	0.294
Diastolic blood pressure (mmHg)	54.049	2	27.024	0.501	0.607
Heart Rate (BPM)	176.602	2	88.301	1.528	0.221
Oxygen Saturation (%)	13.285	2	6.642	4.500	0.013 *

Df, degree of freedom; F, F statistics; *p*, *p*-value;MMHg, millimeters of mercury; BPM, Beats per minute; %, percentage; *, statistically significant (*p* ≤ 0.05)

## Data Availability

Not applicable.

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
