# Peer review of "Variation in Hemodynamic Characteristics during Periodontal Crown-Lengthening Surgical Procedure: An Uncontrolled Cohort Study"

_healthcare, 2022, doi:10.3390/healthcare10050919_

Round 1
Reviewer 1 Report
Authors have attempted to investigate the variation in Hemodynamic Characteristics during Periodontal Crown Lengthening Surgical Procedure. There are a few areas which need clarification.
- How did the authors arrive at the sample size of 44?
- Demographic details of the participants should have been provided.
- Authors should discuss the plausible reasons for the obtained observations
- Authors should include the limitations of the study
- Was the pain perception of the participation evaluated?
Author Response
- How did the authors arrive at the sample size of 44?
Response: Thank you so much for your comment. In this study a convenience sampling technique was used based on the maximum number of patients available and met the inclusion criteria of the study.
- Demographic details of the participants should have been provided.
Response: Demographic details have been added in methodology section as per your comment.
- Authors should discuss the plausible reasons for the obtained observations.
Response: Correction has been done in discussion section as per your comment.
- Authors should include the limitations of the study.
Response: The limitations of the study are included in the discussion section.
- Was the pain perception of the participants evaluated?
Response: Pain perception was not evaluated in this study.
Please see the attachment

Reviewer 2 Report
I find this study is interesting and well written. It is important that clinicians (readers) understand the cardiovascular reactions to dental procedures and do nor overestimate the role of vasoconstrictors.
Author Response
- I find this study is interesting and well written. It is important that clinicians (readers) understand the cardiovascular reactions to dental procedures and do nor overestimate the role of vasoconstrictors.
Response: Thank you so much for your comment. Corrections have been made in the discussion section as per your comment.
Reviewer 3 Report
the study is nice but there is no control group so its conclusion are really weak
different control groups could have been established. just as examples:
1) placebo injection and monitoring of parameters from T1 to T2 to investigate the effects of the anesthesia
2) injection of anesthesia to a group of patients without any further surgical treatment to investigare the effects of surgery.
As the study idea is very nice I encourage the authors to withdraw the submission, repeat the study with a control group and resubmit it again.
Anyway the study could be also published in the present form with the following modifications:
- state in the title this is a case series.
- state in the discussion and in the conclusions that the conclusions are weak due to the absence of a control group
Author Response
Reviewer 3:
Option 1:
- Different injection and monitoring of parameters from T1 to T2 to investigate the effects of the anesthesia
- Injection of anesthesia to a group of patients without any further surgical treatment to investigate the effects of surgery.
As the idea is very nice, I encourage the authors to withdraw the submission, repeat the study with a control group and resubmit it again.
Response: Thank you so much for your kind suggestion. As the study period already over and adding new control group will require new ethical approval and time, we decided to go for the second option you have provided.
Option 2:
The study could be also published in the present form with the following modifications:
- State in the title this is a case series.
Response: Thank you so much for your suggestion. We would like to request if we can keep the same title so that it will be considered as an original article instead of case series, since we have done the study on 44 subjects.
- State in the discussion and in the conclusion that the conclusions are weak due to the absence of a control group.
Response: Thank you so much for your suggestion. We have mentioned about the absence of control group in discussion and conclusion as per your suggestion.
Please see the attachment

Round 2
Reviewer 3 Report
I think you can publish the study but you have to modify the title:
Variation in Hemodynamic Characteristics during Periodontal Crown Lengthening Surgical Procedure: an uncontrolled cohort study.
Author Response
- I think you can publish the study but you have to modify the title.
Response: Thank you so much for your comment. The title has been changed as per the suggestion.
New Title: Variation in Hemodynamic Characteristics during Periodontal Crown Lengthening Surgical Procedure: An Uncontrolled Cohort Study
Please see the attachment
